# GRIN: Generative Relation and Intention Network for Multi-agent Trajectory Prediction

**Longyuan Li**[1]* **Jian Yao**[2]* **Li K. Wenliang**[3]* **Tong He**[4]† **Tianjun Xiao**[4]
**Junchi Yan**[1]† **David Wipf**[4] **Zheng Zhang**[4]

[1]Department of Computer Science and Engineering,
MoE Key Lab of Artificial Intelligence, Shanghai Jiao Tong University
[2]Fudan University
[3]Gatsby Unit, University College London
[4]Amazon Web Services
{jeffli, yanjunchi}@sjtu.edu.cn, kevinli@gatsby.ucl.ac.uk
{htong, tianjux, daviwipf, zhaz}@amazon.com

## Abstract

Learning the distribution of future trajectories conditioned on the past is a crucial problem for understanding multi-agent systems. This is challenging because humans make decisions based on complex social relations and personal intents, resulting in highly complex uncertainties over trajectories. To address this problem, we propose a conditional deep generative model that combines advances in graph neural networks. The prior and recognition model encodes two types of latent codes for each agent: an inter-agent latent code to represent social relations and an intra-agent latent code to represent agent intentions. The decoder is carefully devised to leverage the codes in a disentangled way to predict multi-modal future trajectory distribution. Specifically, a graph attention network built upon inter-agent latent code is used to learn continuous pair-wise relations, and an agent's motion is controlled by its latent intents and its observations of all other agents. Through experiments on both synthetic and real-world datasets, we show that our model outperforms previous work in multiple performance metrics. We also show that our model generates realistic multi-modal trajectories.

## 1 Introduction

Generative trajectory forecasting requires learning the distribution of future trajectories conditioned on the past. It can provide an important foundation of path planning and tracking in many domains [28, 1]. In a highly interactive multi-agent environment, such as team sports, each individual takes actions based on its intentions, relations with, and observation of other agents [7]. For example, an attacking basketball player acts based on his/her strategy along with observations of teammates and the defending team. Similarly, an experienced driver identifies potential dangers from observed actions and motions of other drivers, thereby planning ahead of time [27]. Notably, in many such real-world examples, intentions, relations, and interactions are all dynamic, hidden, and stochastic, leading to highly variable future trajectories.

We argue that to model uncertainty correctly we should account for factors that arise from both individual and social (i.e. relation) perspectives, and do so explicitly. That is to say, individual decision/intention and social relation/observation are inductive biases that should be explicitly

---

*Work completed during internship at AWS Shanghai AI Labs.
†Correspondence authors are Junchi Yan and Tong He.

35th Conference on Neural Information Processing Systems (NeurIPS 2021).

introduced into a model design. Existing works model either intention [15, 32] or relations [24, 3, 36], not both, lead to compromised performance under highly interactive and uncertain environments [22].

To better address the problems above, we propose Generative Relation and Intention Network (GRIN) that explicitly models uncertainties of both the intention and relations, the design comes with an added benefit of being more interpretable. Specifically, our contributions are:

1) We present GRIN, a conditional generative model for multi-agent trajectory prediction, trained by variational inference and learning [35];

2) We propose a disentangled design of the model architecture such that the uncertainty from the intention and relations are modelled separately;

3) We evaluate our model on both a deterministic synthetic dataset [18] and a highly stochastic real-world dataset [41] by single-path and distributional metrics, quantifying the ability to predict accurately while being aware of multiple possibilities. Our model consistently outperforms advanced baselines on all datasets and metrics tested, highlighting the critical advantage of generative modeling of intentions and relations.

## 2 Related Works

Our work is closely related to multi-agent trajectory prediction, relation discovery from trajectory data, and multi-modal prediction, which are discussed in below.

### 2.1 Multi-Agent Trajectory Prediction

Multi-agent trajectory prediction is an actively researched problem due to its broad applications in robot planning [34, 20], traffic prediction [24, 38, 4, 26], sport video analysis [5] and so on. In practical situations, two major observations challenge trajectory prediction: 1) the way agents interact with each other affects the future trajectories, and 2) given very similar past trajectories, the future paths of the same agents may still be very uncertain, producing diverse and multi-modal outcomes. Next, we survey existing works from these two perspectives in detail.

### 2.2 Relation Discovery

Relationship between agents can affect their behaviors through two mechanisms: the *type or state* of the relation and the *communications* among agents contingent on the relations. As such, modeling the interactions between agents using graphs and the message passing mechanism they afford becomes a natural choice. Ivanovic et al. [12] uses heuristics from spatial proximity to get the graph before making predictions. Most other work start from a fully connected graph, then learn the graph edges in a self-supervised way. Kosaraju et al. [19] run graph attention network [39] on the fully connected graph and use the attention weight to decide how much information to share between agents. Kamra et al. [14] developed a dedicated attention mechanism for trajectory prediction from the inductive bias of motion and intents. Instead of using soft-attention, neural relational inference (NRI) proposed by Kipf et al. [18] takes the form of a variational auto-encoder; the edges are modeled as independent, categorical posterior distributions produced by an encoder trained through a biased reparametrization gradient. Jiachen et al. [24] and Graber et al. [8] improved NRI by using graphs that evolve over time. Compared with [24], which is a error-based model with two-stage training pipeline, our model is probabilistic-based model with more customized end-to-end training procedure. [24, 8] compute discrete edge types, which have difficulty modeling continuously varying relationships.

In summary, most work can be considered as RNN-based where the RNN is combined with the graph structure to model both spatial, temporal and inter-agent relations.

### 2.3 Multi-modal Prediction

A distributional estimation of possible trajectories provides more information for downstream tasks, especially for those with risk-sensitive processes. Deep generative models are suitable for such tasks. Most existing works build on the generative adversarial network (GAN) [23, 21, 10] and variational auto-encoder (VAE) [11, 6, 18, 8]. Social-GAN [10] is an early attempt to address this issue. However, it only learns a single mode of behavior with high variance. Social-BiGAT [19] explicitly accounts

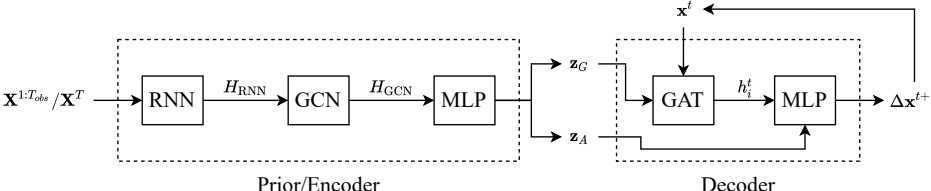

Figure 1: Model architecture of GRIN. It takes in a sequence of $\mathbf{X}^T$ and infers posteriors over the relations in the graph of agents ($\mathbf{z}_G$) and for each individual agent ($\mathbf{z}_A$). Samples of $\mathbf{z}_G$ enter a GAT module to obtain inter-agent features before combining with $\mathbf{z}_A$, the intra-agent features. The GAT computes inter-agent features using the observations as well.

for the multi-modal nature of the prediction problem by forming a reversible transformation between each scene and its latent noise vector. For VAE-based methods, conditional-VAEs [35] provides an appropriate formulation of trajectory prediction, as the past trajectory can be considered as the conditioning variable. The works [11, 34, 33] explore a serials of work using CVAE-based methods. Beyond Gaussian likelihoods, [24] proposes to use a Gaussian mixture as the likelihood to generate multi-modal predictions, which may be helpful given limited flexibility of neural networks.

Previous works on autonomous driving [2, 25, 37] also use graph neural networks to learn the correlation of agents. However, for [37, 2], the relation and intention of agents are entangled into a single latent variable. In [25], it does not involve a probabilistic generative model design to handle multi-modality.

These recent works have mostly considered multi-modality and built mechanisms to handle interaction between agents. However, to our knowledge, none of them disentangle the multi-modality effects from the two sources: inter-agent and intra-agent (or within-agent), which can influence the trajectory in distinct ways. We address this point through a careful architecture design and thorough empirical validation in the rest of the paper.

## 3 Methods

We specify notations, describe the task, and introduce the model architecture and training strategy.

**Notations**   In each scene, we are given $N$ agents across $T$ steps. Denote by $\mathbf{X} \in \mathbb{R}^{N \times T}$ all the trajectory data in this scene, and $x_i^t$ the location of the $i$-th agent at the $t$-th step. Given $T_{obs} \leq T$ as the number of historic frames, we have the observation $\mathbf{X}_P = \mathbf{X}^{1:T_{obs}}$, and the task is to predict $\mathbf{X}_F = \mathbf{X}^{(T_{obs}+1):T}$ based on the observation.

Our motivation is to model the uncertainty of future trajectory given the past and to disentangle the two effects from the intra-agent intention and inter-agent relations, both of which are unobserved. Thus, we model the data by conditional generative model with latent variables $\mathbf{z}_A$ and $\mathbf{z}_G$ that describe agent intention and relational graph, respectively. This model can be trained by maximizing the evidence lower bound (ELBO, [35]), which is usually used as a surrogate for the intractable log-likelihood. During training, when past and future trajectories are available, we infer the latents using an encoder given $\mathbf{X}_P$ and $\mathbf{X}_F$ and optimize the ELBO with SGD. During prediction, we do not require the encoder and predict the future trajectories using the generative model given $\mathbf{X}_P$. Our model, GRIN, is illustrated in Figure 1. We describe the details in the following subsections.

### 3.1  Encoder

The encoder computes an approximate posterior $q(\mathbf{z}|\mathbf{X}_P, \mathbf{X}_F)$ with samples $\mathbf{z} = [\mathbf{z}_A, \mathbf{z}_G]$ conditioned on $\mathbf{X}_P$ and $\mathbf{X}_F$. To allow disentanglement between the two types of latents, we explore an inductive bias built into our decoder architecture described in Section 3.3.

To start, we concatenate $\mathbf{X}_P$ and $\mathbf{X}_F$ along the time dimension and use a recurrent neural network (RNN) to compute feature of the full-length trajectory of each agent:

$$\mathbf{H}_{\text{RNN}}^E = \text{RNN}([\mathbf{X}_P, \mathbf{X}_F]), \tag{1}$$

where $\mathbf{H}_{RNN}^E \in \mathbb{R}^{N \times M_h}$, where $N$ is the the number of agents and $M_h$ is the feature dimension. Next, we assume a complete graph $G$ for the $N$ agents, and further improve the feature via a graph convolutional network (GCN):

$$\mathbf{H}_{\text{GCN}}^E = \text{GCN}(G, \mathbf{H}_{\text{RNN}}^E), \tag{2}$$

where $\mathbf{H}_{\text{GCN}}^E \in \mathbb{R}^{N \times M_h}$. Assuming $\mathbf{z}$ follows a multi-variate Gaussian distribution $\mathcal{N}(\boldsymbol{\mu}, \mathbf{I}\boldsymbol{\sigma})$, with the reparameterization trick [31, 16], we sample $\boldsymbol{\epsilon} \sim \mathcal{N}(\mathbf{0}, \mathbf{I})$, and obtain $\mathbf{z} = \boldsymbol{\epsilon} \times \boldsymbol{\sigma} + \boldsymbol{\mu}$, where the $\boldsymbol{\mu}$ and $\boldsymbol{\sigma}$ are predicted by Multi-Layer Perceptrons (MLP):

$$\begin{aligned} \boldsymbol{\mu}^E &= \text{F}_\mu(\mathbf{H}_{\text{GCN}}^E) \\ \boldsymbol{\sigma}^E &= \text{F}_\sigma(\mathbf{H}_{\text{GCN}}^E) \end{aligned} \tag{3}$$

The encoder's outputs are $\mathbf{z}_A \in \mathbb{R}^{N \times M_Z}$, $\mathbf{z}_G \in \mathbb{R}^{N \times M_Z}$ where $M_Z$ is the dimension of $\mathbf{z}_A, \mathbf{z}_G$.

## 3.2 Prior

The prior $p(\mathbf{z}|\mathbf{X}_P)$ characterize the conditional distribution of $\mathbf{z}$ given only $\mathbf{X}_P$. In our implementation, the prior shares the model weights with the encoder. In this case, the encoder and prior are only different in terms of the input data to the RNN in Eq 1. By weight-sharing, the same RNN sees inputs with different lengths, and the KL-divergence term in the loss enforces the encoder and decoder to extract similar information from the input. Therefore, the RNN is encouraged to learn to encode the same information from $\mathbf{X}_P$ and $[\mathbf{X}_P, \mathbf{X}_F]$. Besides, weight-sharing reduces its memory cost.

$$\begin{aligned} \mathbf{H}_{\text{RNN}}^P &= \text{RNN}(\mathbf{X}_P) \\ \mathbf{H}_{\text{GCN}}^P &= \text{GCN}(\mathbf{G}, \mathbf{H}_{RNN}^P) \\ \boldsymbol{\mu}^P &= \text{F}_\mu(\mathbf{H}_{\text{GCN}}^P) \\ \boldsymbol{\sigma}^P &= \text{F}_\sigma(\mathbf{H}_{\text{GCN}}^P) \end{aligned} \tag{4}$$

## 3.3 Decoder

The decoder $p(\mathbf{X}_F|\mathbf{z}_A, \mathbf{z}_G, \mathbf{X}_P)$ produces future trajectories conditioned on the random variable $\mathbf{z}$ and the past trajectory $\mathbf{X}_P$. Following [18], we assume that that latent $\mathbf{z}$ is a sufficient summary of the distance history and define the likelihood as $p(\mathbf{X}^{t+1}|\mathbf{z}, \mathbf{X}^{1:t}) = p(\mathbf{X}^{t+1}|\mathbf{z}, \mathbf{X}^t)$. Therefore, given $\mathbf{z}$ and $\mathbf{X}^t$, the decoder can predict $\hat{\mathbf{X}}^{t+1}$. For $t > T_{obs}$, the input to the decoder is its own prediction $\hat{\mathbf{X}}^t$.

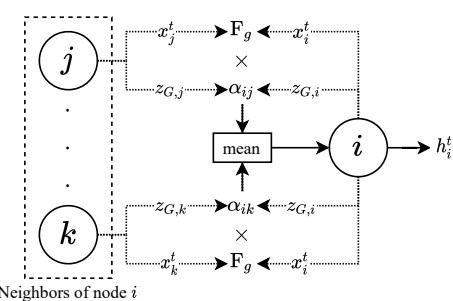

Neighbors of node $i$

Figure 2: The modified GAT module.

The decoder is built with a simple inductive bias that is nonetheless important for capturing inter-agent relations, which we describe now. In many practical situations, the relation between two agent largely depend on the agents themselves and is thus correlated. For example, for three particles A, B and C each with either positive or negative charge, if one knows that A attracts B, and B attracts C, then A must repels C. Thus, the posterior over the relations are highly correlated. Such a correlation over the relations can be more difficult to characterize, since dimensionality of relations scales as $N^2$. Thus, we propose to model the relations as a function of each pair of agent, rather than independent as assumed in previous work, e.g. [18], which yields correlated uncertainty over the relations.

To incorporate this intuition, we assume a complete graph for the $N$ agents, and apply a modified Graph Attention Network (GAT) module [39] to describe their pairwise relations via the attention mechanism. Specifically, we use $\mathbf{z}_G$ to compute the attention weight on edges and update node features with observation $\mathbf{X}^t$. This module is illustrated in Figure 2 which implements:

$$\begin{aligned} e_{ij} &= \langle \text{F}_d(z_{G,i}) \cdot \text{F}_s(z_{G,j}) \rangle \\ \alpha_{ij} &= \text{softmax}_j\{e_{ij}\} \end{aligned} \tag{5}$$

where $\langle \cdot \rangle$ is the dot product, $\text{F}_s$ and $\text{F}_d$ are two MLP layers for source ($z_{G,j}$) and destination ($z_{G,i}$) node features respectively, and $\alpha_{ij}$ is the edge attention weight for the later graph convolution

computation. Thus, the edges become correlated through the graph latents $\mathbf{z}_G$ of each individual. By definition, the attention thus computed is not constrained to be symmetric but contains the set of symmetric relationships. When the true relationship is indeed symmetric, such as the force between charged particles, we expect this attention matrix to by almost symmetric.

Further, one agent may have its own intention that is usually private and not available to the others. The agent-specific latent $\mathbf{z}_A$ models such private intentions and does not enter the GAT module. Altogether, it is likely that our model disentangles an agent's own intention ($\mathbf{z}_A$) and the relations through ($\mathbf{z}_G$), which we carefully validate in experiments.

The rest of the decoder follows similar from previous work. Given $\alpha_{ij}$ and $\mathbf{X}^t$, we perform one round of message passing to extract the feature for each agent:

$$h_i^t = \frac{1}{N-1} \sum_j \alpha_{ij} \mathbf{F}_g([x_i^t, x_j^t]) \tag{6}$$

Next, we incorporate $h_i^t$ with $\mathbf{z}_{A,i}$ to predict the $\Delta x_i^{t+1} = x_i^{t+1} - x_i^t$. We assume $\Delta \hat{x}_i^t$ follows a Gaussian distribution $\mathcal{N}(\Delta x_i^t, \sigma_x)$, where $\sigma_x$ is for the entire dataset. Then $\sigma_x$ is estimated by a single learned parameter $\hat{\sigma}_x$, and the mean is predicted by:

$$\Delta \hat{x}_i^{t+1} = \mathbf{F}_x([h_i^t, z_{A,i}]) \tag{7}$$

The GAT module models the relationships among the agents given $\mathbf{z}_G$ and their location $\mathbf{X}^t$.

## 3.4 Training

For training, we sample $\mathbf{z} = [\mathbf{z}_A, \mathbf{z}_G]$ from the encoder, $q(\mathbf{z}|\mathbf{X}_P, \mathbf{X}_F)$, and then optimize the ELBO:

$$\text{ELBO}_{\mathbf{z}} = -\mathbb{E}[\log(p(\mathbf{X}_F|\mathbf{z}, \mathbf{X}_P))] + \text{KL}[q(\mathbf{z}|\mathbf{X}_P, \mathbf{X}_F)||p(\mathbf{z}|\mathbf{X}_P)]. \tag{8}$$

Since our decoder predicts $[\Delta \hat{x}_i^t, \hat{\sigma}_x]$ for agent $i$ at step $t$, and it follows a Gaussian distribution. We approximate the expectation with the sample mean:

$$-\mathbb{E}[\log(p(\mathbf{X}_F|z, \mathbf{X}_P))] \approx \frac{1}{\mathcal{Z}} \sum_{t=T_{obs}+1}^{T} \sum_{i=1}^{N} \left[ \frac{n}{2} \log(2\pi \hat{\sigma}_x^2) + \frac{\sum (\Delta \hat{x}_i^t - \Delta x_i^t)^2}{2\hat{\sigma}_x^2} \right] \tag{9}$$

where $\mathcal{Z}$ is the normalization term, and $n$ is the dimension of $\Delta x_i^t$. Since the outputs of prior and the encoder are Gaussian, the second term can be computed analytically:

$$\text{KL}[q(\mathbf{z}|\mathbf{X}_P, \mathbf{X}_F)||p(\mathbf{z}|\mathbf{X}_P)] = \text{KL}[\mathcal{N}(\boldsymbol{\mu}^E, \mathbf{I}\boldsymbol{\sigma}^E)||\mathcal{N}(\boldsymbol{\mu}^P, \mathbf{I}\boldsymbol{\sigma}^P)]. \tag{10}$$

Following [10], we sample multiple $\mathbf{z}$'s for one batch, and only compute the gradient from the lowest $\text{ELBO}_{\mathbf{z}}$:

$$\mathcal{L} = \min_{\mathbf{z}} \{\text{ELBO}_{\mathbf{z}}\} \tag{11}$$

## 3.5 Trajectory Generation

To generate future trajectories, we first sample $\mathbf{z}$ from the prior, $p(\mathbf{z}|\mathbf{X}_P)$, and then roll out the prediction from the decoder for any $t \geq T_{obs}$:

$$\begin{aligned} \Delta \hat{\mathbf{x}}_i^{t+1} &= \text{Decoder}(\mathbf{z}, \mathbf{X}^t) \\ \hat{\mathbf{x}}_i^{t+1} &= \mathbf{x}_i^t + \Delta \hat{\mathbf{x}}_i^{t+1} \end{aligned} \tag{12}$$

The first input of roll out is the ground truth observation $\mathbf{X}^{T_{obs}}$. For $t > T_{obs}$, we use $\hat{\mathbf{X}}^t$ as the input.

## 4 Experiments

In this section, we analyze the performance of GRIN against several baselines on two datasets. First, we introduce the datasets, baselines, and the evaluation metrics. Next, we show quantitative results comparing GRIN with baselines and ablation study on GRIN. Finally, we conduct qualitative case-study of GRIN's behavior on the trajectory prediction task.[3]

---

[3]Code and data are available at `https://github.com/longyuanli/GRIN_NeurIPS21`.

### 4.1 Datasets

**Charged Dataset [18].** We experiment with a simulated deterministic systems: charged particles, which controlled by simple physics rules. In each scene, there are 5 charged particles. Each particle has either a positive or negative charge with equal probability. Particles with the same charge repel each other, and vise versa. We set $T = 100$ and $T_{obs} = 80$ for each scene. We generate 50K scenes for training, and 10K each for validation and test respectively.

**NBA Dataset [41].** This dataset contains tracking data from the 2012-2013 NBA season, and is accessible in [30]. In this dataset, each trajectory contains the 2D positions of the basketball and the players from both sides. Each team is consisted of 5 players. We preprocess the data such that each scene has 50 frames and spans approximately 8 seconds of play, and the first 40 frames are historic, i.e. $T = 50$ and $T_{obs} = 40$.

### 4.2 Evaluation Metrics

The potential uncertainty and multi-modality in the trajectory datasets makes the evaluation difficult. Instead of just measuring the point-wise absolute match between prediction and ground truth, we check both point-wise accuracy and distributional distance by the following different metrics.

**1) Average Displacement Error (ADE) and Final Displacement Error (FDE).** Average Displacement Error is the L2 distance between the ground truth and predicted trajectories. Final Displacement Error measures the L2 distance between the ground truth final destination and predicted final destination. For stochastic models, we report the best-of-100 displacement error of each trajectory/destination.

**2) Conditional Maximum Mean Discrepancy (MMD).** MMD [9] is a non-parametric measure of distributional distance between two distributions. Here, our goal is to learn a conditional distribution of the future path given the past, so we used the conditional version defined in [13]. See Appendix for details.

**3) Negative Log Likelihood (NLL).** With the test dataset, we compute ELBO, an upper bound on the NLL, on the predicted trajectories. Since the computation of ELBO asks for a sample of the random variables for each trajectory, we compute 100 ELBO values and report the mean and the standard deviation. We only report this metric on models that are trained with ELBO.

### 4.3 Baselines

To make the comparison complete, we decouple the designs to several aspects and compare with the representatives. While there are methods that integrate tracking with a advanced perceptual backbone [19, 33], here we mainly validate our approach of modeling inter- and intra-agent uncertainty, while the integration with other components is also worth future investigation.

**Instance Interaction.** Some existing works take the form of RNN plus interaction module at each timestamp. *Fuzzy Query Attention (FQA)* [14] is a recent work following this paradigm, which models the inductive bias from relative motion, intent and interaction.

**Inferred Latent Graph.** Works in this paradigm take the form of a variational auto-encoder, infer the latent graph from past trajectory. Message passing on the decoder is conducted on the inferred graph. Representative works include *Neural Relational Inference (NRI)* [18] and its dynamic version *Dynamic Neural Relational Inference (dNRI)* [8].

**Generative Model.** Generative model is a suitable tool to handle multi-modality and uncertainty. One classic work following this line is *Social-GAN* [10], as it is a GAN-based method. It can be considered as a general building block, and can be extended by introducing visual input.

**Implementation Details.** We implement our model using PyTorch [29], which has BSD-style license. The graph neural network modules are implemented using DGL [40], which has Apache 2.0 license. We train the model with the Adam [17] optimizer with the learning rate of $0.001$. For all datasets, we set the dimension of $\mathbf{z_G}$ and $\mathbf{z_A}$ to be 2, the hidden dimension of all MLPs in the encoder and decoder to be 128. The batch size is set to be 128 for charged particle dataset and 64 for NBA dataset. Following common practice in VAE training, we let the predicted variance $\hat{\sigma}_x$ be a parameter that is trained along with all other parameters in the model. We estimate the total amount of computation is

Table 1: Performance comparison with baseline models. MMD with $p < 0.05$ are marked with *. Deterministic models are marked with †.

| Model | Charged | | NBA | | | |
|---|---|---|---|---|---|---|
| | ADE | FDE | ADE | FDE | $\text{MMD}^2_{\times 10^3}$ | NLL |
| NRI [18] | 0.63 | 1.30 | 2.10 | 4.56 | 1.11 | 586.9±0.0 |
| dNRI [8] | 0.94 | 1.93 | 2.02 | 4.52 | 1.49* | 662.6±0.0 |
| FQA† [14] | 0.82 | 1.76 | 2.42 | 4.81 | 1.38* | - |
| Social-GAN [10] | 0.66 | 1.25 | 1.88 | 3.64 | 1.18* | - |
| GRIN (Ours) | **0.52** | **1.09** | **1.72** | **3.59** | **1.04** | **507.5±0.2** |

Table 2: Disentanglement ablation results of the proposed GRIN on NBA dataset. MMD with $p < 0.05$ are marked with *. Deterministic models are marked with †.

| Model | ADE | FDE | $\text{MMD}^2_{\times 10^3}$ |
|---|---|---|---|
| Deterministic† | 2.03 | 4.26 | 1.13 |
| Deterministic $\mathbf{z}_G$ | 1.89 | 3.85 | 1.12 |
| Deterministic $\mathbf{z}_A$ | 1.87 | 3.87 | 1.17* |
| Full Model | **1.72** | **3.59** | **1.04** |

in total around 500 hours on an AWS p3.8xlarge EC2 instance with 4 Nvidia V100 GPU. The full details on the architecture and baseline implementations are discussed in the Appendix.

## 4.4 Quantitative Evaluation

**Baseline Comparison.** We quantitatively compare our proposed model against the others. In Table 1, we compare on Charged and NBA datasets. Since there's no multi-modality in Charged dataset, we only report ADE and FED. Since FQA is deterministic, and Social-GAN is not optimized via ELBO, we do not compute NLL for these two models. GRIN achieves the best or on par performance under all the settings. We run hypothesis test of MMD scores and the difference is statistically significant between our prediction and the predictions from dNRI, FQA and Social-GAN. Although the ADE and FDE performance of Social-GAN is similar to GRIN, their MMD is significantly less optimal than ours. This indicates GRIN learns a better distribution of future trajectory.

**Disentanglement Ablation Study.** We examine the quantitative impact of $\mathbf{z}_G$ and $\mathbf{z}_A$ in the model. We make $\mathbf{z}_G$ or $\mathbf{z}_A$ deterministic by only predicting the mean as a deterministic latent vector, and remove the KL penalty of this term. Under this setting, the baselines are not computing a valid ELBO thus we don't report the NLL score. The experiments are conducted on NBA dataset, and the results are available in Table 2. Obviously, the deterministic model is less capable of making predictions in an uncertain environment. The two models with one fixed $\mathbf{z}$ both give compromised performance comparing to the full model. Especially for a deterministic $\mathbf{z}_A$, only sampling the $\mathbf{z}_G$ the MMD is significantly larger than the full model. The results shows that having $\mathbf{z}_G$ and $\mathbf{z}_A$ together improves on both the discrepancy error and the distribution distance. We also provide perturbation experiments of $\mathbf{z}_A$ and $\mathbf{z}_G$ to show the disentanglement of latent space, which can be found in the Appendix.

## 4.5 Qualitative Analysis with Visualization

We illustrate the predicted trajectories for scenes for both datasets. The figures support that GRIN can predict trajectories accurately in the deterministic environment, and can also capture uncertainty in the stochastic environments. Note that we just illustrate a few number of examples due to space limit, and more can be found in Supplementary to show that the observations is general.

**Interactive Scene in a Deterministic Environment.** On Charged dataset, we manually design a scenario that some particles start with close trajectories, thus it is highly interactive. Moreover, although we don't have neutral particles in the training and test environment, a neutral particle with no charge is inserted into the scene. The prediction from GRIN is illustrated in Figure 3. We observe

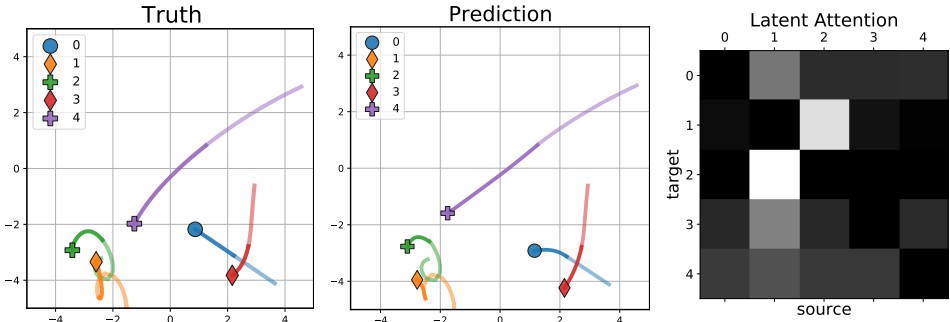

Figure 3: Trajectory prediction on charged particles dataset. The left two figures are ground truth and predicted trajectories. Diamond, plus and circle marks negative-charged, positive-charged and neutral particles respectively. The right plot is the learned edge attention $\alpha_{i,j}$, where the $i$-th row records the attention weights from the $i$-th node's neighbors, where the cell on row $i$ and column $j$ records the attention weights from the $j$-th node to the $i$-th node. Higher attention scores are in lighter color.

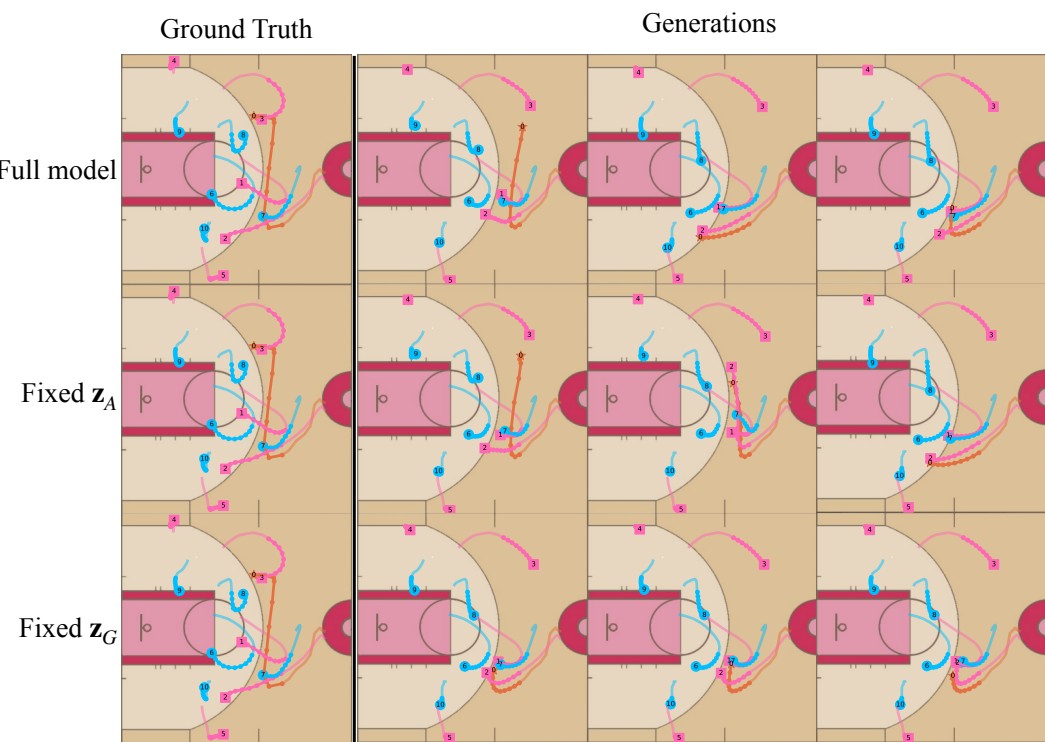

Figure 4: Samples of generated trajectories on NBA dataset. For each trajectory, the light solid line is the past trajectory, and the dotted line is the future trajectory. Attacking players are pink, defending players are blue, and the basketball is brown. The first column is the ground truth trajectories, and the following three are the generated samples. The first row shows the generated trajectories from the full model, that we sample from both $\mathbf{z}_G$ and $\mathbf{z}_A$. The second row shows the generated trajectories from the model that only samples different $\mathbf{z}_G$ with a fixed sample of $\mathbf{z}_A$, and the third contains the results from a fixed sample of $\mathbf{z}_G$ and different samples of $\mathbf{z}_A$.

that for a closed pair of particles (1 and 2), our model accurately captures the attracting forces as illustrated in the attention matrix. Interestingly, for the neutral particle 0, the model only picks up a minor effect from particle 1, and it doesn't impact the other particles either. This empirically proves that our attention module is capable to capture interpretable relationships among agents, and ignores the less relevant pairs.

**Uncertainty and Multi-modality in Real-world Environment.** We visually show the multi-modality in the real-world NBA dataset. First, to show that our model is capable to capture the uncertainty, we illustrate a few scenarios where the prediction trajectories are highly diverged in Figure 4. To further inspect the effect from $\mathbf{z}_G$ and $\mathbf{z}_A$ separately, we sample $\mathbf{z}_G$ repeatedly with one fixed random sample of $\mathbf{z}_A$ and vise versa, then plot the trajectories under the same scenario. From the ground truth, we see that player 2 carries the ball in the past, and he passes the ball to player 3. From the generation in the first two rows, we see that player 2 could either pass the ball or carry it himself. In the third row, the model only predicts trajectories with a smaller variance. We argue that this case demonstrates that $\mathbf{z}_G$ has a more significant impact on the uncertainty of trajectories. This observation aligns with the common sense of basketball, as it is highly interactive and players' movements are mostly affected by the inter-agent relationships.

## 5 Conclusion and Discussions

oIn this paper, we present GRIN for multi-agent trajectory prediction. Our model achieves state-of-the-art performance on synthetic and real-world datasets across multiple metrics. Moreover, GRIN explicitly disentangles the sources of future uncertainty in the decoder, and produces convincing interpretable outputs in real-world scenarios.

### 5.1 Outlook and Future Works

Based on this work, we aim at several meaningful directions as future works.

**1) Vision-based Trajectory Prediction.** GRIN takes trajectories as inputs directly. While for unstructured vision data which is an important application scenario for our approach, e.g. video, an end-to-end pipeline is often needed to perform tracking and extract inter- and intra-agent information from video data to feed into our model, as done in [19, 33].

**2) Dynamic Trajectory Prediction.** Real-world environment is highly interactive and dynamic. This can be done by extending our model to continuously update $\mathbf{z}_A$ and $\mathbf{z}_G$ as time rolls forward.

**3) Sparse Relationship Modeling.** For simplicity, we have used a fully connected graph to model agent interactions. Real-world environments often induce sparse connectivity, so connectivity can change over time. Leveraging sparsity dynamically can potentially improve efficiency and scalability.

### 5.2 Limitations

Our approach may still have limitations in modeling and predicting very complex multi-agent systems. Although it captures the correlations in the relations, we cannot guarantee that the relations are consistent with the rules of the context (e.g. laws of physics between binary charged particles). One can either build in a particular decoder to represent such constraints or use a surrogate model that learns and approximate the rules.

### 5.3 Potential Negative Social Impacts

Trajectory prediction can be privacy-sensitive when the targets are human individuals. Our model may further mine the underlying interactions and even relations from the observed behaviors which raise some concerns.

## Acknowledgement

Longyuan Li and Junchi Yan were partly supported by National Key Research and Development Program of China (2020AAA0107600), Shanghai Municipal Science and Technology Major Project (2021SHZDZX0102), and NSFC (61972250, 72061127003).

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
