# Supplementary Material for GRIN: Generative Relation and Intention Network for Multi-agent Trajectory Prediction

**Longyuan Li**[1*]   **Jian Yao**[2*]   **Li K. Wenliang**[3*]   **Tong He**[4†]   **Tianjun Xiao**[4]
**Junchi Yan**[1†]   **David Wipf**[4]   **Zheng Zhang**[4]

[1]Department of Computer Science and Engineering,
MoE Key Lab of Artificial Intelligence, Shanghai Jiao Tong University
[2]Fudan University
[3]Gatsby Unit, University College London
[4]Amazon Web Services
{jeffli, yanjunchi}@sjtu.edu.cn, kevinli@gatsby.ucl.ac.uk
{htong, tianjux, daviwipf, zhaz}@amazon.com

## A  Quantitative Evaluation on Pedestrian Dataset

In addition to the datasets reported in the main text, we also evaluate our model on the publicly-available ETH [12]/UCY [11] pedestrian dataset, which consists 5 sets of data from 4 unique scenes. There has been several settings to evaluate on this dataset in the literature. Gupta et al. [5] and Ivanovic et al. [6] adopt a leave-one-out approach, training on 4 sets and testing on the remaining set. On the other hand, Kamra et al. [8] merges the 5 sets and splits out train/validation/test sets from the merged set. In evaluation, the model observes 8 frames and predicts 12 frames. In this work, we only evaluate the performance of the predicted 12 frames on the pedestrians that appear throughout the whole 20 frames to avoid the absence of ground-truth due to occlusion or out-of-camera-view. We sample 100 times for each given history, select the best one to compute ADE and FDE only.

Several scene transferring issues are brought up by recent papers on the leave-one-out approach. To name a few, the majority of the trajectories in the *Hotel* set have different directions with the ones in the other 4 sets. Schöller et al. [13] claim that rotation augmentation is required to make the learned model valid to at least beat the linear baseline model. Zhang et al. [15] notice that the *ETH* set has a different sampling frame rate with the others. Due to these issues, data augmentation plays an important role to get good performance on this dataset. Typical augmentation methods include rotation, adjusting frame rate, training with various trajectory lengths, etc. In this paper, we use the same evaluation setting as *FQA* [8] such that no scene transferring issue need to be taken care of. We do not apply augmentation for all methods in Table 1 except for *Trajectron* since augmentation is deeply coupled with *Trajectron* training code. Our model can be combined with these augmentation techniques, but for now we mark that as a future work alongside with adding visual feature and extending to dynamic model. We compare GRIN with *Social-GAN* [5] and *Trajectron* [6] baselines. GRIN produces better results on best-of-100 ADE/FDE performance compared with the baselines without augmentation. Our ADE performance is slightly worse than *Trajectron* [6] while our FDE is on-par, despite that *Trajectron* [6] is trained with variable sequence lengths and other augmentation methods. Meanwhile, our best-of-20 ADE/FDE performance is better than all baselines, and the gap between best-of-20 and best-of-100 is the lowest, meaning our model is more certain on the predictions.

---

[*]Work completed during internship at AWS Shanghai AI Labs.
[†]Correspondence authors are Junchi Yan and Tong He.

35th Conference on Neural Information Processing Systems (NeurIPS 2021).

Table 1: Quantitative performance comparison on ETH/UCY pedestrian trajectory dataset

| Model | NBA | | ETH/UCY | |
|---|---|---|---|---|
| | ADE 20/100 | FDE 20/100 | ADE 20/100 | FDE 20/100 |
| NRI [10] | 2.12/2.10 | 4.67/4.56 | 0.17/0.17 | 0.38/0.38 |
| FQA† [8] | 2.42/2.42 | 4.81/4.81 | 0.19/0.19 | 0.44/0.44 |
| Trajectron [6] | 2.82/2.31 | 5.65/4.64 | **0.10/0.07** | **0.21/0.15** |
| Social-GAN [5] | 1.95/1.88 | 3.81/3.64 | 0.12/0.10 | 0.24/0.20 |
| GRIN | **1.75/1.72** | **3.68/3.59** | **0.10**/0.09 | **0.21/0.15** |

## B  Visualization Examples on NBA Dataset

We have shown in our results that our model captures better the distribution of predictions compared with baselines, suggesting that it may be able to capture multimodality in complex scenes. In this section, we visualize and analyze four more randomly picked scenes and predictions from the NBA dataset in Figure 1. The first column contains the ground truth trajectories for each scene, and the other columns are different predictions given each trajectory history. Although picked randomly, these scenes cover a range of diverse situations on the court.

First, we focus on the scene in the third row. The third row shows obvious multi-modality of player 4 (pink square next to the star in ground truth; numbers are visible in zoom-up) and the ball (marked by the star). From the generated results in the third row, we see that player 4 either carries the ball in hand (second and third columns) or passes the ball to the teammate player 3 (fourth and fifth columns). Although in the ground truth the ball is not passed, one can argue that the pass is also a reasonable move, since player 4 faces multiple opponents while the view from player 4 to 3 is open and clear.

The first, second, and fourth rows show less uncertain scenarios. There are generated trajectories that don't match perfectly with the ground truth. For instance, player 5 in the first scene and player 8 in the second scene. However, these cases include sudden changes of direction, and such changes are more challenging to infer from the past trajectory. Overall, we qualitatively demonstrate that the predictions align with the common sense of basketball.

## C  Model Architectures and Hyper-parameters

We provide the hyper-parameters of our model and baseline models in this section. All our experiments were done on an AWS p3.8xlarge EC2 instance with 4 Nvidia V100 GPU, with Ubuntu 18.04 system. Neural networks were implemented using PyTorch framework v1.8.1+cu102, and graph neural networks include graph attention neural networks and graph convolution networks are defined using DGL [14] v0.6.1. The code for implementing GRIN can be found at the supplementary material file. For all models, the input dimension is 4 (2 for positional coordinates plus 2 for velocity vector). For baseline models that have multiple design choices, we try the combination of choices and select the best one based on the performance on the validation dataset. Due to the different scalability of models, we did not specifically tune the batch size but adjusted it to fill up the GPU memory. All models are trained with Adam optimizer [9] with learning rate $1 \times 10^{-3}$. For *NRI* and *dNRI*, the teacher forcing steps is set to 10 for NBA dataset, 20 for charged dataset and 4 for ETH/UCY dataset. Different from the original paper of *NRI* and *dNRI*, where the variance of the output distribution is set to a fixed value $5 \times 10^{-5}$, we let it be a learned parameter so that the model adapts to different datasets automatically. We found that this can improve the NLL performance on NBA dataset. For ETH/UCY dataset, we sample 20 times for the latent variables during training and only back-propagate for the one getting the smallest loss in order to encourage diversity as described in *Social-GAN* [5].

### C.1  Neural Relational Inference (NRI)

We adapt the code from official repository (`https://github.com/ethanfetaya/NRI`) of the paper [10]. We set the number of edge types to 2, and the first edge type is not skipped in the decoder

Ground Truth                                    Generations

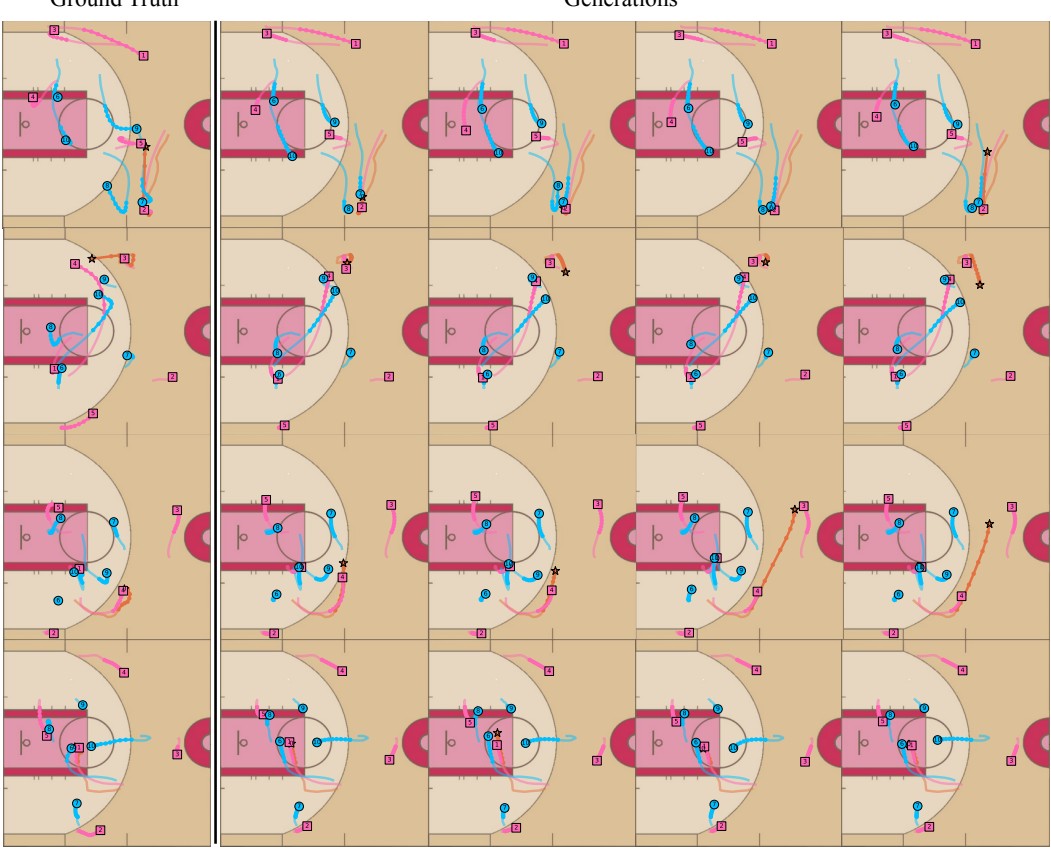

Figure 1: More visualization samples of generated trajectories on NBA. For each trajectory, the light solid line is the past trajectory, and the dotted line is the future trajectory. Attacking players are pink, defending players are blue, and the basketball is brown. Each row corresponds to a single movement history of a scene. The first column is the ground truth trajectories, and the other columns are the generated predictions.

(since skip the first type gave worse performance). We tried combinations of CNN/MLP encoder and RNN/MLP decoder with different sizes of hidden layers. We found that CNN encoder with MLP decoder with hidden dimension 256 and no dropout works best on both charged dataset and NBA dataset.

## C.2  Dynamic Neural Relational Inference (dNRI)

We adapt the code from the official repository (`https://github.com/cgraber/cvpr_dNRI`) of the paper [3]. Similar to the implementation of NRI, we set the number of edge types to 2, and the first edge type is not skipped in the decoder. We use RNN encoder and MLP decoder with hidden dimension 256 and no dropout.

## C.3  Fuzzy Query Attention (FQA)

We use the code from the official repository (`https://github.com/nitinkamra1992/FQA`) of the paper [8]. As suggested by the paper, we use 8 query-key pair for all datasets. The dimension for keys and queries is set to 4, and the dimension for yes-no responses is set to 6. The hidden state dimension of the shared LSTM is set to 32.

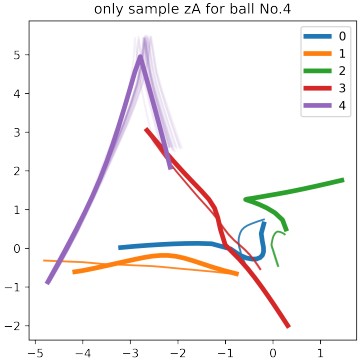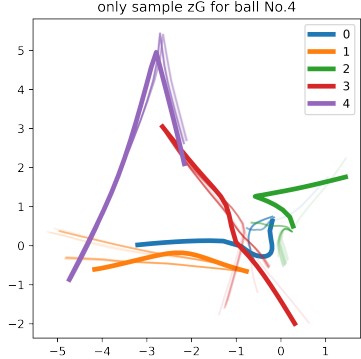

(a) Sample $\mathbf{z}_{A,4}$ of particle No.4 (purple) while all other $\mathbf{z}$s are fixed

(b) Sample $\mathbf{z}_{A,G}$ of particle No.4 (purple) while all other $\mathbf{z}$s are fixed

Figure 2: Visualization of ablation study on the latent disentanglement by fixing all $\mathbf{z}$ except for $\mathbf{z}_A$ or $\mathbf{z}_G$ of particle No.4 (purple), the thick solid lines are ground-truth trajectories, and thin semi-transparent lines are generated trajectories from samples.

### C.4 Social GAN

We use the code from the official repository (`https://github.com/agrimgupta92/sgan`) of the paper [5]. The RNN module is LSTM. The hidden dimension is 32 for the encoder generator, 64 for the encoder discriminator and 32 for the decoder. For NBA and charged ball dataset, we sample 10 times for the latent variable in training. For ETH/UCY dataset, we sample 20 times in training.

## D Qualitative Analysis of Disentanglement

To demonstrate the effects of $\mathbf{z}_A$ and $\mathbf{z}_G$, we perform a pair of perturbation experiments on the Charged dataset:

- We fix $\mathbf{z}_{A,0}, \ldots, \mathbf{z}_{A,4}$ for all five particles and $\mathbf{z}_{G,0}, \ldots, \mathbf{z}_{G,3}$ for four of the five particles, and only sample $\mathbf{z}_{G,4}$ for one particle, then observe the distribution of the generated trajectories.
- We fix $\mathbf{z}_{G,0}, \ldots, \mathbf{z}_{G,4}$ for all five particles and $\mathbf{z}_{A,0}, \ldots, \mathbf{z}_{A,3}$ for four of the five particles, and only sample $\mathbf{z}_{A,4}$ for one particle, then observe the distribution of the generated trajectories.

In this pair of experiments, we observe:

- As shown in Figure. 2(a), when sampling only for $\mathbf{z}_{A,4}$ one particle, only its own generated trajectory is affected, and all other four particles' trajectories are not affected. This clearly shows the disentanglement of inter- and intra-agent factors.
- As shown in Figure. 2(b), when sampling only $\mathbf{z}_{G,4}$ for one particle, the generated trajectories of all five particles are affected.

## E Ablation Studies

In the experiment we use a very small $\mathbf{z}$ dimension, where z_dim is 2. We also provide results with larger $\mathbf{z}$ dimension on the NBA dataset. As shown in Table 2, z_dim is the dimension of a $\mathbf{z}_A$ or $\mathbf{z}_G$ vector for each agent; ADE, FDE, and NLL are the metrics on the test data. From the table we can tell that the dimension of $\mathbf{z}_A$ and $\mathbf{z}_G$ has minimal impact on ADE and FDE, and a large dimension (=20 in our ablation) slightly improves the NLL.

We also did an ablation study with only a single $\mathbf{z}_A$ or $\mathbf{z}_G$ as follows:

| z dim | ADE | FDE | NLL |
|---|---|---|---|
| 2 | 1.715 | 3.598 | 507.495 |
| 5 | 1.736 | 3.636 | 509.381 |
| 10 | 1.732 | 3.614 | 508.102 |
| 20 | **1.71** | **3.594** | **503.393** |

Table 2: Ablation study of $\mathbf{z}_A$ and $\mathbf{z}_G$ dimension on the NBA dataset.

| model | ADE | FDE | NLL |
|---|---|---|---|
| $\mathbf{z}_A$ only | 1.95 | 3.98 | 531.2 |
| $\mathbf{z}_G$ only | 1.88 | 3.83 | 530.5 |
| $\mathbf{z}_A$ & $\mathbf{z}_G$ | **1.69** | **3.52** | **504.9** |

Table 3: Ablation study of using single $\mathbf{z}_A$ or $\mathbf{z}_G$ on the NBA dataset.

- remove $\mathbf{z}_A$ and double the size of $\mathbf{z}_G$.
- remove $\mathbf{z}_G$, double the size of $\mathbf{z}_A$ and compute the attention weights using the hidden states.

For models with single $\mathbf{z}_A$ or $\mathbf{z}_G$, we set the dimension of $z$ to be 20. For $\mathbf{z}_A$ & $\mathbf{z}_G$ model, we set the dimension of each $z$ to be 10. The experiments is conducted with different seeds from ablation study of z dimension, such that the results are slightly different from Table 2. From Table 3, we can see that the performance of the model is worst in the case of only $\mathbf{z}_A$ and reaches its best in the case of both $\mathbf{z}_A$ and $\mathbf{z}_G$, showing that $\mathbf{z}_A$ and $\mathbf{z}_G$ encode different implicit states in a disentangled way.

## F  Maximum Mean Discrepancy

In our task, the goal is to learn a good conditional density of future trajectory given the past. As the distribution of futures may be multi-modal, the naive mean squared error (MSE) measure may be misleading. Also, the MSE ignores the correlations between time steps and between agents and only measures the marginal distributions. For a simple 1-D example, assume that the correct future paths given some history is either to the left by 2 units (-1, -2) or to the right by 2 units (+1, +2), each with equal probability, then a model that predicts (-1,+2) and (1,-2) with equal probability will on average have the same MSE with another model that predicts (-1,-2) and (+1,+2) with equal probability. Clearly, the second model has the same distribution with the truth and should be preferred, but MSE cannot tell any difference. Such difference can be identified by a distributional metric.

We choose to measure distributional distance using a well-established metric known as the maximum mean discrepancy (MMD, [4]). In this section, we aim to give a brief summary of MMD, its application to model comparison and extension to conditional MMD. MMD measures the difference between two distributions based on the difference between their expectations of a "witness" function $f$ in some space of functions $\mathcal{H}$.

$$\text{MMD}[P\|Q] := \sup_{f \in \mathcal{H}: \|f\|_{\mathcal{H}} \leq 1} |\mathbb{E}_P[f] - \mathbb{E}_Q[f]|.$$

Please see [4] for detail; we only provide an intuitive description. Essentially, the best witness function $f$ is one that can tell apart $P$ from $Q$ while satisfying a smoothness constraint $\|f\|_{\mathcal{H}} \leq 1$. It is similar to the KL divergence in that they both compute distributional distances. When the space of functions is the reproducing kernel Hilbert space associated with kernel function $k(\cdot, \cdot)$ satisfying

$$k(x, y) = \phi(x) \cdot \phi(y)$$

for canonical feature vectors $\phi$ that can be infinite-dimensional. Then the MMD can be estimated using samples, as it is equivalent to

$$\text{MMD}[P\|Q]^2 = \|\mathbb{E}_{X \sim P}[\phi(X)] - \mathbb{E}_{Y \sim Q}[\phi(Y)]\|_{\mathcal{H}}^2 \tag{1}$$

$$= \mathbb{E}_{X,X' \sim P}[k(X, X')] + \mathbb{E}_{Y,Y' \sim Q}[k(Y, Y')] - 2\mathbb{E}_{Y \sim Q, Y \sim P}[k(X, Y)] \tag{2}$$

If we interpret the kernel as a form of similarity measure, then this metric makes intuitive sense. It computes the average similarity from intra-distribution samples from $P$ and $Q$ (first two term), then minus twice the average similarity between inter-distribution samples (last term). In practise, care

must be taken to avoid biasing the estimate when computing the within-distribution similarities: we must only compare the kernel using distinct samples and avoid valuating the kernel using the same sample. Therefore, given finite samples $\{x_i\}_{i=1}^N \sim P$ and $\{x_j\}_{j=1}^M \sim Q$, an empirical unbiased estimate of the MMD is

$$\widehat{\text{MMD}}^2[P\|Q] = \tfrac{1}{N(N-1)} \sum_{i=1}^N \sum_{i'\neq i}^N k(x_i, x_{i'}) + \tfrac{1}{M(M-1)} \sum_{j=1}^M \sum_{j'\neq j}^M k(y_j, y_{j'}) - \tfrac{2}{MN} \sum_{i=1}^N \sum_{j=1}^M k(x_i, y_j) \tag{3}$$

This distributional distance can be used to test whether two sets of samples $p$ and $q$ are from the same distribution, and also compare if $\text{MMD}[R\|P] < \text{MMD}[R\|Q]$ give a third distributions $R$ [1]. In short, if $R \neq P$ and $R \neq Q$, then the $\widehat{\text{MMD}}^2[R\|P] - \widehat{\text{MMD}}^2[R\|Q]$ using samples has an asymptotic normal distribution, using which we could determine whether the difference is significantly smaller or greater than zero.

For the purpose of measure predictive distribution, we need to adapt the MMD for testing conditional distributions. We suppose that the real data distribution is $R$ from which we split samples along the time dimension as $[H, Z] \sim R$, where $H$ is the history and $R$ is the real future. In addition, we have a model trained to predict the future $X$ given the past $H$, resulting in a model distributions $P(X|H)$. Here, the variables $X, Y$ and $Z$ are defined over the same domain but may follow different distributions.

Following a similar definition in [7], we define the following conditional MMD (using minimal technical terminology):

$$\text{CMMD}^2[R\|P] = \|\mathbb{E}_{H\sim R}[L(\cdot, H)\zeta(H)]\|_{\mathcal{H}}^2, \tag{4}$$
$$\zeta(H) := \mathbb{E}_{Z\sim R(Z|H)}[\phi(Z)] - \mathbb{E}_{X\sim P(X|H)}[\phi(Y)], \tag{5}$$

where $L(\cdot, \cdot)$ is a operator (vector-valued kernel) such that

$$L(x, y) = l(x, y)I,$$

where $l$ is another kernel similar to $k$, and $I$ is the identity mapping in $\mathcal{H}$. One can think of $L$ as acting on each element of the difference vector $\zeta$. Thus, to compute the CMMD, we first compute a difference between the moments of $\phi$ under two models; this difference is then "smoothed" using an operator $L$ weighted by the distribution of the history $R$. The squared CMMD is then the norm of this smoothed and weighted difference.

By the reproducing property of the kernels, the CMMD reduces to an MMD over $R(H)R(Z|H)$ and $R(H)P(Z|H)$

$$\text{CMMD}^2[R\|P] = \|\mathbb{E}_{[H,Z]\sim R}[L(\cdot, H)\phi(Z)] - \mathbb{E}_{[H,X]\sim R(H)P(Y|H)}[L(\cdot, H)\phi(Y)]\|_{\mathcal{H}}^2 \tag{6}$$
$$= \mathbb{E}_{[H,Z],[H',Z']\sim R}[l(H, H')k(Z, Z')] + \mathbb{E}_{[H,X],[X',Z']\sim RP}[l(H, H')k(X, X')] \tag{7}$$
$$- \mathbb{E}_{[H,Z]\sim R,[H',X']\sim RP}[l(H, H')k(Z, X')]. \tag{8}$$

And the similar test in [1] can be used for comparing with a second model that has learned $Q(Z|H)$. In our experiments, We used the IMQ kernel for $l$ and $k$ [2]

$$k(x, y) = \left(1 + \left\|\frac{x-y}{\sigma}\right\|_2^2\right)^{-0.5}$$

where $\sigma$ is determined by the median distance between pairs of the history (for $l$) and true future (for $k$). This is to ensure that the CMMDs computed on all models are comparable to each other for a given test dataset.

We modified the code from [1] to carry out the CMMD test for model comparison using the product kernel $l(\cdot, \cdot)k(\cdot, \cdot)$ as above. In addition, we implemented the data splitting procedure described by [7] to avoid biased estimates. That is, we split the test data into two halves $[H_1, Z_1]$ and $[H_2, Z_2]$, then use each baseline model to predict the future given $H_2$, giving $[H_2, X] \sim RP$ for model $P$. We also predict using GRIN, giving $[H_2, Y] \sim RQ$. For each baseline model $P$, we test the null hypothesis that

$$\widehat{\text{CMMD}}[R\|P] < \widehat{\text{CMMD}}[R\|Q],$$

using a significance level of 0.05.

Specifically for this work, we computed the MMD by drawing a single prediction given a history in the test dataset. More predicted samples could be incorporated in principle but would take longer to run, and the CMMDs are unlikely to be drastically affected (the test results may change due to potentially smaller variance in the estimates). Finally, since MMD compares distributions supported on spaces with fixed dimensionality, we cannot apply to datasets that have a variable number of people, and thus do not report it for the ETH/UCY pedestrian trajectory dataset.