# OpenReview forum: "GRIN: Generative Relation and Intention Network for Multi-agent Trajectory Prediction"
_NeurIPS.cc/2021/Conference — NeurIPS 2021 Poster_

### Official Review · Reviewer_C2Xv · 2021-07-08

**Rating:** 7
**Confidence:** 4

**Summary:**

This paper develops a multivariable multimodal trajectory prediction model titled GRIN which is based on a CVAE framework. Specifically, future trajectories are conditioned on two latent vectors: one (the “interaction” latent) which is intended to represent the interactions between all pairs of entities, and one (the “intent” latent) which is intended to represent the “intent” of each individual entity. These meanings are enforced by the model architecture, as the interaction latent is used to compute attention values which are used to aggregate hidden representations for all pairs of variables, while the intent latent only impacts the prediction of its corresponding entity. Experiments are run on a few trajectory prediction datasets; GRIN outperforms other approaches which model interactions using latent variables.


**Limitations And Societal Impact:**

The limitations described are somewhat general and vague, and would apply to most (if not all) of the baselines as well. Are there any limitations to this approach that are unique to it? Are there observed scenarios where the model tends to fail?

The raised negative societal impacts are valid, and there are no ethical concerns raised by this work in its current form.

**Main Review:**

### Originality:
The novel component of this paper is the model architecture, as the CVAE formulation for trajectory prediction has been developed/used in prior work. The idea of interpreting a per-entity latent variable as an intent is not novel (e.g. [A]), nor is the idea of modeling relations between agents as latent variables (e.g. [7, 17]). To my knowledge, the combination of the two employed here is novel.

### Quality:
- The data used for comparison and the baseline models chosen are appropriate.
- The novelty of this approach regards the use of the different types of latent variables; however, the predicted latent variables and what they encode are not explored in much detail. The only visualization provided is in Figure 3, for the synthetic data. It would be additionally interesting to explore whether the latent variables encoded any meaningful information for the other datasets.
- This model uses very small latent vector sizes (2 per latent “type”, for a total of 4 dimensions). No experiments are presented which investigate how the size of these vectors indicate performance.
- The appendix presents experimental results on additional data (ETH/UCY), where an additional baseline (Trajectron) is used for comparison. This model performs comparably on this dataset; it would be useful to understand how Trajectron performs on the other data as well (does it still perform comparably? Better? Worse?)
- The latent variable sampling procedure described in lines 168/169 are somewhat strange to use in this context, since the ELBO contains expectations which are computed over the latent variables ([9] doesn’t run into this issue, since it uses a GAN loss). No experiments are run which investigate this choice - does doing this “min sampling” improve performance?


### Clarity:
The approach is presented clearly, and builds sensibly on prior work related to multimodal trajectory prediction and interaction modeling. Code is provided, and the paper is detailed enough to ensure reproducibility.

### Significance:
The model outperforms related approaches which model interactions between entities using latent variables. It is unclear whether this approach could be considered “state of the art” compared to other multimodal trajectory forecasting approaches (as mentioned previously, the experimental results indicate it performs comparably to other approaches). It is unclear whether the predicted latent variables are meaningful beyond leading to improved performance, as only one set of latent variables for one trajectory for one dataset are visualized.


### Final Thoughts:
Though GRIN performs well compared against related approaches, it is not completely clear why a practitioner should use this model over alternatives. There is not enough evidence presented to call this approach "state-of-the-art," and there is not enough evidence presented that the predicted latent vectors are useful for interpretation purposes. If the goal is to present a model which explicitly models interactions between entities (such as NRI or dNRI) and performs better compared to prior approaches which do so, then more comparisons need to be made to better understand which component of the model leads to improved performance and whether the latent variable representation leads to meaningful interpretation.


[A]Salzmann, Tim, Boris Ivanovic, Punarjay Chakravarty, and Marco Pavone. "Trajectron++: Dynamically-feasible trajectory forecasting with heterogeneous data." ECCV 2020.



# POST-REBUTTAL UPDATE
Thanks for the comment. It's great to see you were able to fix a few things and get better results. Additionally, the additional ablations regarding the impact of the different latent variables on predictions are a valuable addition to the paper. I'm increasing my score based on these additions.

**Time Spent Reviewing:**

2.5 hours

---

> ### Author Response · Authors · 2021-08-10
> **Response to Reviewer C2Xv**
>
> We thank the reviewer for the detailed review as well as the suggestions for improvement. Our responses to the reviewer’s comments are below. We appreciate the reviewer’s positive comment on data and baseline model choices.
>
> **To my knowledge, the combination of the two employed here is novel.**
>
> Thank the reviewer for acknowledging our novelty. We think our method of constructing the interaction is also new and valuable: we treat interaction as arising from each individual, and thus can model continuous relationships and correlations over the interactions. Previous works [7,17] treat the edges as conditionally independent categorical variables, which may result in graphs that do not respect certain physical constraints.
>
> **It would be additionally interesting to explore whether the latent variables encoded any meaningful information for the other datasets.**
>
> This is a valuable suggestion. Interpreting interactions in real applications is difficult and likely requires an interdisciplinary effort and domain expertise. In the future, we hope to collaborate with scientists and practitioners to further examine the information in these latents in certain scientific research or practical applications.
>
> **This model uses very small latent vector sizes (2 per latent "type", for a total of 4 dimensions). No experiments are presented which investigate how the size of these vectors indicate performance.**
>
> Here we present the results from the ablation study on the dimension of $z_A$ and $z_G$.
>
> | z dim 	| ADE   	| FDE   	| NLL    	|
> |-------	|-------	|-------	|---------	|
> | 2     	| 1.715 	| 3.598 	| 507.495 	|
> | 5     	| 1.736 	| 3.636 	| 509.381 	|
> | 10    	| 1.732 	| 3.614 	| 508.102 	|
> | 20    	| **1.71**  	| **3.594** 	| **503.393** 	|
>
> In this table, "z dim" is the dimension of a $z_A$ or $z_G$ vector for each agent; ADE, FDE, and NLL are the metrics on the test datar.
>
> From the table we can tell that the dimension of $z_A$ and $z_G$ has minimal impact on ADE and FDE, and a large dimension (=20 in our ablation) slightly improves the NLL.
>
> **This model performs comparably on this dataset; it would be useful to understand how Trajectron performs on the other data as well (does it still perform comparably? Better? Worse?)**
>
> We have updated the Trajectron results on NBA and Charged datasets in the tables below. GRIN still outperforms all baselines in these two datasets. Trajectron reports good performance on pedestrian datasets in its paper, while we observe its poor performance on more interaction-intensive datasets such as NBA and Charged Particles where scene-level consistency affects the performance. The decoder design for Trajectron can’t guarantee such scene-level consistency.
>
> Charged
>
> |    Model        		|     ADE     	|     FDE     	|
> |-------------------	|:-------------:	|:-------------:	|
> |     Trajectron    	|     0.90    	|     1.77    	|
> |     NRI           		|     0.63    	|     1.30    	|
> |     dNRI          	|     0.94    	|     1.93    	|
> |     FQA           	|     0.82    	|     1.76    	|
> |     Social-GAN    	|     0.66    	|     1.25    	|
> |     GRIN(Ours)    	|     **0.52**    	|     **1.09**    	|
>
>
> NBA
>
> |    Model        		|     ADE     	|     FDE     	|
> |-------------------	|:-------------:	|:-------------:	|
> |     Trajectron    	|     2.31    	|     4.64    	|
> |     NRI           		|     2.10    	|     4.56    	|
> |     dNRI          	|     2.02    	|     4.52    	|
> |     FQA           	|     2.42    	|     4.81    	|
> |     Social-GAN    	|     1.88    	|     3.64   	|
> |     GRIN(Ours)    	|     **1.69**    	|     **3.52**    	|
>
>
> **The latent variable sampling procedure described in lines 168/169 are somewhat strange to use in this context, since the ELBO contains expectations which are computed over the latent variables ([9] doesn’t run into this issue, since it uses a GAN loss). No experiments are run which investigate this choice - does doing this "min sampling" improve performance?**
>
> Indeed we are not optimizing the ELBO directly. We introduce this modification following [9]. We observe that the test NLL is similar with or without this trick, while test ADE and FDE are improved with this trick.
>
> **It is unclear whether this approach could be considered "state of the art" compared to other multimodal trajectory forecasting approaches (as mentioned previously, the experimental results indicate it performs comparably to other approaches)**
>
> For the results in our paper, we had a significant MMD test (marked by *) favoring our model in Table 1. In contrast to ADE and FDE, it compares the joint predictive distributions of two models with the true data distribution. Further details on MMD are discussed in the supplementary and cited work [8].
>
> We have now obtained substantially better results than reported in the table. The improvement comes from these factors:
>
> 1. We fixed a few bugs that unfairly limited GRIN's performance in our submission, one example is that we didn't shuffle the data samples during the training of GRIN. Other baseline methods are not affected by these bugs as we train them with their official implementations.
> 2. Inspired by the reviewer's suggestion on the dimension of $z_A$ and $z_G$, we increase the dimension of $z_A$ and $z_G$ from 2 to 20.
>
> Taking NBA dataset as an example:
>
> |    NBA          	| ADE   	| FDE   	|
> |--------------	|-------	|-------	|
> | Social-GAN      	| 1.88 	| 3.64 	|
> | GRIN             | 1.83      | 3.66       |
> | +bug fix         | 1.72      | 3.59       |
> | +z dimension     | *1.69* 	| *3.52* 	|
>
> **It is unclear whether the predicted latent variables are meaningful beyond leading to improved performance**
>
> To further demonstrate the effect of $z_G$ and $z_A$, we perform a pair of ablation experiments on the Charged dataset:
> 1. We fix $z_{A,0},\dots,z_{A,4}$ for all five particles and $z_{G,0},\dots,z_{G,3}$ for four of the five particles, and only sample the $z_{G,4}$ for one particle, then observe the generated trajectory.
> 2. We fix the $z_{G,0},\dots,z_{G,4}$ for all five particles and $z_{A,0},\dots,z_{A,3}$ for four of the five particles, and only sample the $z_{G,4}$ for one particle, then observe the generated trajectory.
>
> In this pair of experiments, we observe:
> 1. When sampling $z_{G,4}$ for one particle, the generated trajectories of all five particles are affected.
> 2. When sampling $z_{A,4}$ for one particle, only its own generated trajectory is affected, and all other four particles' trajectories are not affected.
>
> Please check the visualization through these anonymous links:
> 1. https://ibb.co/HpSP6gh
> 2. https://ibb.co/TWtN2xm
>
> In the visualization, the solid lines are ground-truth trajectories, and semi-transparent lines are generated trajectories from samples.
>
> **If the goal is to present a model which explicitly models interactions between entities (such as NRI or dNRI) and performs better compared to prior approaches which do so, then more comparisons need to be made to better understand which component of the model leads to improved performance and whether the latent variable representation leads to meaningful interpretation.**
>
> We conducted an ablation study to distinguish the contribution of different latents, which is now extended with results presented in our responses to reviewers 8qCr and WhRr. Please also see responses to the previous comment on interpreting the latent variables. These interpretations validated our model in a simple situation. However, interpreting interactions in real applications is difficult and likely requires an interdisciplinary effort and domain expertise. We hope to collaborate with scientists and practitioners to further examine the representation of these interactions in research and applications.
>
> **The limitations described are somewhat general and vague, and would apply to most (if not all) of the baselines as well. Are there any limitations to this approach that are unique to it? Are there observed scenarios where the model tends to fail?**
>
> We will analyze failure cases to ensure a complete characterization of GRIN’s performance. The model treats each agent equally while, in reality, some agents can behave very differently. For example, in the NBA dataset, the basketball is treated as an agent just like the players, even though it can travel much faster than players and follow a deterministic trajectory when passed between players. The predicted ball trajectory during a pass sometimes shows unrealistic curvature that violates the law of physics.

---

> > ### Comment · Reviewer_C2Xv · 2021-08-21
> > **Increased my score**
> >
> > Thanks for the response - I have edited my initial review based on it.

---

### Official Review · Reviewer_WhRr · 2021-07-11

**Rating:** 6
**Confidence:** 5

**Summary:**

This work presents a deep VAE model for the multi-agent trajectory prediction problem. The assumption that social relationship between agents and each intention is a bias that should be introduced into the network explicitly.

**Limitations And Societal Impact:**

Yes

**Main Review:**

I think the paper is moderately clear in terms of the content display and the proposed idea is original. The ablation study between Z_A and Z_G was helpful to support the claim of the inductive bias.

- This raises a question, what if Z_A and Z_G are treated as one tensor Z_AG?
- Also, I didn't quite understand what makes Z_A agent only and Z_G graph only? is it by design? More details on this are appreciated.
- One main concern: In eq(1) X_P and X_F are concatenated in the encoder. Practically, you would only use the X_P in the encoder, isn't this like introducing the target to be predicted into the network? How does this works while testing if this is an only training mechanism?
- Other works such as Social-STGCNN, EvolveGraph, Trajectron++ and similar lines of works do model each agent as a graph and define the relationship between them through the means of graph edged. They define it either by direct handmade function or by learning such a relation. Also, in Trajectron++ an agent graph is created (similar to Z_G). Where does GRAIN stands between these line of works, especially the concept of Z_A and Z_G seems to exists in previous lines of works in a sense?
- The Qualitative analysis misses failure cases and why it happens.

=+=+=+=+=+=+=+=+=+ POST REBUTTAL =+=+=+=+=+=+=+=+=+

I believe that all the concerns where answered in a proper manner except one point. This point is the ADE/FDE 100 samples on ETH/UCY datasets in which most of the papers use the 20 samples metric and rarely uses the 100 samples metrics. Sharing the same concern as pxYX reviewer. Yet, I will raise my score. I believe posting the 20 samples ADE/FDE will improve the paper.


**Time Spent Reviewing:**

1

---

> ### Author Response · Authors · 2021-08-10
> **Response to Reviewer WhRr**
>
> We thank the reviewer for the detailed review as well as the suggestions for improvement. Our responses to the reviewer’s comments are below.
>
> **What if $z_A$ and $z_G$ are treated as one tensor $z_{AG}$?**
>
> Then we will no longer have disentanglement but gain a potentially more flexible model to capture the statistical relationships in the data. The model may show a better fit to the data, such as a higher ELBO, but we can no longer have access to inter- and intra-agent representations. Further, for certain data (e.g. NBA) where a separate representation could make sense, our treatment provides a good inductive bias. We ran additional experiments to test the performance of this idea, and indeed separating these two produced better results. In the table "single $z$" means the setting with only one $z$ vector, and the dimension of this $z$ vector is equivalent to the dimension of $z_G$ or $z_A$.
>
> Notice that the numbers for GRIN in the table below are better than the numbers in Table 1 of our paper submission. The improvement comes from these factors:
> 1. We fixed a few bugs that unfairly limited GRIN's performance in our submission, one example is that we didn't shuffle the data samples during the training of GRIN.
> 2. Inspired by reviewer C2Xv on the dimension of $z_A$ and $z_G$, we increase the dimension of $z_A$ and $z_G$ from 2 to 20.
>
>
> |    NBA          	| ADE   	| FDE   	| NLL     	|
> |--------------	|-------	|-------	|---------	|
> | single $z$		| 1.72 	| 3.62 	| 509.2 	|
> | $z_G$ \& $z_A$	| **1.69** 	| **3.52** 	| **504.9** 	|
>
>
> |   Charged | ADE   	| FDE   	|
> |--------------	|:-------:	|:-------:	|
> | single $z$		| 0.536 	| 1.137 	|
> | $z_G$ \& $z_A$	| **0.52** 	| **1.09** 	|
>
>
> **I didn't quite understand what makes $z_A$ agent only and $z_G$ graph only? is it by design?**
>
> Yes, it is by design. They are separated primarily through the decoder architecture: $z_G$ goes through a graph attention mechanism to capture the interactions between different agents, and the resulting message $h$ is sent to Eqn(7). In contrast, $z_{A,j}$ enters directly into our Eqn (7) and determines the trajectory of agent $j$; it is combined with the summarized (averaged) message of all other agents $h_i^t$ but never interacts with individual trajectories of other agents.
>
> To further demonstrate the effect of $z_G$ and $z_A$, we perform a pair of ablation experiments on the Charged dataset:
> 1. We fix $z_{A,0},\dots,z_{A,4}$ for all five particles and $z_{G,0},\dots,z_{G,3}$ for four of the five particles, and only sample the $z_{G,4}$ for one particle, then observe the generated trajectory.
> 2. We fix the $z_{G,0},\dots,z_{G,4}$ for all five particles and $z_{A,0},\dots,z_{A,3}$ for four of the five particles, and only sample the $z_{G,4}$ for one particle, then observe the generated trajectory.
>
> In this pair of the experiment, we observe:
> 1. When sampling $z_{G,4}$ for one particle, the generated trajectories of all five particles are affected.
> 2. When sampling $z_{A,4}$ for one particle, only its own generated trajectory is affected, and all other four particles' trajectories are not affected.
>
> Please check the visualization through these anonymous links:
> 1. https://ibb.co/HpSP6gh
> 2. https://ibb.co/TWtN2xm
>
> In the visualization, the thick solid lines are ground-truth trajectories, and thin semi-transparent lines are generated trajectories from samples.
>
> **Practically, you would only use the $X_P$ in the encoder, isn't this like introducing the target to be predicted into the network? How does this work while testing if this is an only training mechanism?**
>
> Just to clarify, we use the standard training method for conditional variational auto-encoder models, for example, "Learning Structured Output Representation using Deep Conditional Generative Models" (NeurIPS 2015). The encoder sees the past $X_P$ as the condition and the future $X_F$ as the data to reconstruct, while the prior only sees the past $X_P$ without $X_F$. We apologize for the typo in Eqn 4: there's no $X_F$ in the input to RNN of the prior.
>
> During training when the future $X_F$ is provided as targets, the encoder takes this to compute the posterior $p(z|X_F,X_P)$. This posterior helps us to train the generative model that describes the conditional density $p(X_F|X_P)$. At the same time, the KL divergence in the loss helps to close the gap between the encoder and the prior.
>
> During testing, we pass $X_P$ to prior in order to sample $z$, and only use the decoder to compute $p(X_F|X_P)$ where the latent variable is conditioned on $X_F$ and marginalized out.
>
> **Where does GRIN stands between these line of works, especially the concept of $z_A$ and $z_G$ seems to exists in previous lines of works in a sense**
>
> Our work is distinct from others in the following ways:
> 1. We formalize the notion of inter- and intra-agent elements as latent variables in a probabilistic generative model with well-defined priors over these latents and likelihoods. This allowed us to
>    - Model simultaneously but disentangled inter- and intra-agent effects (Social-STGCNN does not separate, most other work considers only one of the two.) The disentanglement is achieved by the inductive bias in the GRIN decoder. The tensors sampled from the inter-agent variables are used to specify an attention weight that weighs the ongoing message passing within the decoder.  In this way, the prediction for a node is influenced both by the $z_G$ sampled on its own node and the $z_G$ sampled on other nodes interacting with it. The prediction will then maintain scene-level consistency. (Trajectron++ samples on each node, the prediction only relies on the value sampled on its own node, and no further message passing is done after sampling. Scene consistency is not guaranteed.)
>    - Train the model using the principled VAE framework without handcrafting loss functions, separate stages, or tuning their weightings. (EvolveGraph has two separate stages of training with custom schedules.)
>    - Predict multimodal future paths as enabled by the stochastic latent states (Social-STGCNN focuses on the arguably highly uni-modal prediction tasks, as implied in their Conclusions).
> 2. Our interaction graph is computed based on agent-specific latent variables, rather than discrete edge types computed from an encoder in most other work (e.g., NRI). The advantage of our approach is discussed in lines 132-140. In short, it can capture correlations among edges which are usually regarded as (conditionally) independent.
> 3. We employed a complete set of performance metrics, and notably the MMD metric which is new in this field. Unlike ADE which treats predictions at different times steps independently, MMD is sensitive to the joint predictive distribution over time. In addition, it enables a principled statistical test to compare between models.
>
> **The Qualitative analysis misses failure cases and why it happens.**
>
> We will analyze failure cases to ensure a complete characterization of GRIN’s performance. The model treats each agent equally while, in reality, some agents can behave very differently. For example, in the NBA dataset, the basketball is treated as an agent just like the players, even though it can travel much faster than players and follow a deterministic trajectory when passed between players. The predicted ball trajectory during a pass sometimes shows unrealistic curvature that violates the law of physics.

---

> > ### Comment · Reviewer_WhRr · 2021-08-21
> > **Thanks for your response**
> >
> > Dear,
> > I looked into all reviewers comments and your responses. I believe that all the concerns where answered in a proper manner except one point. This point is the ADE/FDE 100 samples on ETH/UCY datasets in which most of the papers use the 20 samples metric and rarely uses the 100 samples metrics. Sharing the same concern as pxYX reviewer. Yet, I will raise my score. I believe posting the 20 samples ADE/FDE will improve the paper.

---

> > > ### Author Response · Authors · 2021-08-25
> > > **Follow up response to reviewer WhRr**
> > >
> > > We thank the reviewer for the suggestion. We present the results of the suggested experiment in the table below:
> > >
> > > | Model      	| NBA        	|            	| ETH        	|            	|
> > > |------------	|------------	|------------	|------------	|------------	|
> > > |            	| ADE 20/100 	| FDE 20/100 	| ADE 20/100 	| FDE 20/100 	|
> > > | GRIN       	| **1.75**/**1.69**  	| **3.68**/**3.52**  	| **0.10**/0.09  	| **0.21**/**0.15**  	|
> > > | Social-GAN 	| 1.95/1.88  	| 3.81/3.64  	| 0.12/0.10  	| 0.24/0.20  	|
> > > | Trajectron 	| 2.82/2.31  	| 5.65/4.64  	| **0.10**/**0.07**  	| **0.21**/**0.15**  	|
> > >
> > > This table shows that:
> > >
> > > - For the NBA dataset, GRIN outperforms the other two baselines under both best-of-20 and best-of-100 settings;
> > > - For the ETH dataset, GRIN performs similarly to Trajectron, and both are better than Social-GAN.
> > >
> > > In addition, we would like to point out that in the paper of Trajectron, they also used the setting of best-of-100 in their main quantitative comparison experiments. Together with the above results, we hope the concerns regarding the best-of-N setting could be eased.

---

### Official Review · Reviewer_8qCr · 2021-07-16

**Rating:** 6
**Confidence:** 3

**Summary:**

This work proposes GRIN, an architecture that explicitly models the latent uncertainty of social and individual aspects of future prediction of multi-agent systems. The architecture is trained using a conditional variational inference framework, and uses graph attention networks to allow for social attention during the future generation stage. The architecture is evaluated on a Charged particles dataset, an NBA motion forecasting dataset and pedestrian datasets (appendix). Quantitative and qualitative results show that the proposed approach is better than the baselines, and showcases the utility of disentangling the latent space into social and individual components.

**Limitations And Societal Impact:**

Limitations and social impacts are adequately discussed in the conclusion.

**Main Review:**

Strengths:
- The major innovation in this work is the decomposition of the latent space into two parts: the first addresses inter-agent relationships and the second addresses intra-agent intention. The effectiveness of this approach is shown both quantitatively (Tab. 2) and qualitatively (Fig. 4).
- The introduction, related work and method sections are well-written and clearly communicate the message to the reader.

Weaknesses:
- The related work section does not address recent motion forecasting approaches, some of which use a similar message passing mechanism [1,2, 3]. These methods have been explored in the context of autonomous driving.
- Other than disentangling the latent space, I am not entirely sure if the architecture as a whole is novel. The architecture seems quite similar to the one proposed in [1].
- The quantitative improvement in Table 1 over Social-GAN seems marginal. Given that there are more recent approaches that outperform Social-GAN on motion forecasting tasks, it would've been interesting to compare to those methods (e.g., [4]).
- The disentanglement Ablation study performed (line 242 and Table 2) seems somewhat counter-intuitive. It seems to me that the correct ablation would be to 1) remove $z_A$ and double the size of $z_G$; 2) remove $z_G$, double the size of $z_A$ and compute the attention weights using the hidden states $h_i^t$. This would allow one to more accurately assess the effect of disentangling the latent space.

References:
- [1] Casas, Sergio, et al. "Implicit latent variable model for scene-consistent motion forecasting." Computer Vision–ECCV 2020: 16th European Conference, Glasgow, UK, August 23–28, 2020, Proceedings, Part XXIII 16. Springer International Publishing, 2020.
- [2] Li, Lingyun Luke, et al. "End-to-end contextual perception and prediction with interaction transformer." 2020 IEEE/RSJ International Conference on Intelligent Robots and Systems (IROS). IEEE, 2020.
- [3] Tang, Charlie, and Russ R. Salakhutdinov. "Multiple futures prediction." Advances in Neural Information Processing Systems 32 (2019): 15424-15434.
- [4] Sadeghian, Amir, et al. "Sophie: An attentive gan for predicting paths compliant to social and physical constraints." Proceedings of the IEEE/CVF Conference on Computer Vision and Pattern Recognition. 2019.

Clarifications:
- Figure 4: The color of the ball (brown) is very difficult to distinguish given the background of the court.
- The explanation of Figure 4 at line 266 can be improved by outlining that with a fixed $z_G$ the ball always remains with agent 2. It's not entirely clear what happens when we fix $z_A$.
- For the NBA dataset, is there a special identifier for the ball to differentiate it from players in the encoding? If not, how does the model know to treat it's movement differently?

Minor corrections:
- References 18 and 19 are the same.


**Time Spent Reviewing:**

4

---

> ### Author Response · Authors · 2021-08-10
> **Response to Reviewer 8qCr**
>
> We thank the reviewer for the detailed review as well as the suggestions for improvement. Our responses to the reviewer’s comments are below.
>
> **The related work section does not address recent motion forecasting approaches, some of which use a similar message passing mechanism [1, 2, 3]**
>
> Thanks for the suggestions. Those works are related to our method. However, our model is different from them in several ways. The major difference in our method is the disentanglement design. We formalize the notion of inter- and intra-agent elements as latent variables in a probabilistic generative model with well-defined priors over these latents and likelihoods. This allowed us to model simultaneously but disentangled inter- and intra-agent effects. This design is suitable for scenarios where the interactions are intensive while agents also have the freedom to make their own decisions. [1, 2, 3] haven’t modeled both. Besides this major difference:
> 1. Compared with [1] and [3], our decoder utilizes the sampled  $z_G$ to explicitly specify an attention $\alpha$ that weights the ongoing interaction between agents. The advantage of our approach is discussed in lines 132-140. While for the decoder in [1, 3], no such inductive bias is introduced.
> 2.[2] does not contain a probabilistic latent variable generative model design to handle multi-modality.
>
> **Other than disentangling the latent space, I am not entirely sure if the architecture as a whole is novel. The architecture seems quite similar to the one proposed in [1]**
>
> One of the novel contributions is the interaction graph constructed by attention, so each edge is determined by the properties of the connected agents, and the variability of the edge is correlated rather than independent. This also allows the edge to take on continuous values. The work referenced by the reviewer does not partition the inter- and intra-agent factors, so the architecture is in fact very different. Another notable difference is that our model follows the CVAE framework, while their decoder is not probabilistic and so is not optimized by the ELBO.
>
>
> **The quantitative improvement in Table 1 over Social-GAN seems marginal. Given that there are more recent approaches that outperform Social-GAN on motion forecasting tasks, it would've been interesting to compare to those methods (e.g., [4]).**
>
> The improvement is in fact quite substantial and statistically significant based on the MMD test: GRIN’s predictions that are statistically significantly closer to the data than baselines are marked with a * in Table 1. We have now obtained substantially better results than reported in the table. The improvement comes from these factors:
>
> 1. We fixed a few bugs that unfairly limited GRIN's performance in our submission, one example is that we didn't shuffle the data samples during the training of GRIN. Other baseline methods are not affected by these bugs as we train them with their official implementations.
> 2. Inspired by reviewer C2Xv on the dimension of $z_A$ and $z_G$, we increase the dimension of $z_A$ and $z_G$ from 2 to 20. Taking NBA dataset as an example:
>
> |    NBA          	| ADE   	| FDE   	|
> |--------------	|-------	|-------	|
> | Social-GAN      	| 1.88 	| 3.64 	|
> | GRIN             | 1.83      | 3.66       |
> | +bug fix         | 1.72      | 3.59       |
> | +$z$ dimension     | **1.69** 	| **3.52** 	|
>
> SoPhie [4] falls into a conditional GAN framework, uses several attention neural networks for processing but does not contain interpretable latent variables. The component most related to ours is the social attention; it relies on a particular feature difference computed according to the sorted spatial distance between agents (Eqn. (3) in [4]). This is not appropriate for more complex scenes where the distance between agents is not indicative of the strength of their interactions (e.g. team sports).
>
> **It seems to me that the correct ablation would be to 1) remove $z_A$ and double the size of $z_G$; 2) remove $z_G$, double the size of $z_A$ and compute the attention weights using the hidden states**
>
> The reviewer is correct that we should control the latent size in ablation. The new results are below, and using both types of latent variables drastically improves over using either one alone.
>
> |    NBA          	| ADE   	| FDE   	| NLL     	|
> |--------------	|-------	|-------	|---------	|
> | $z_A$ only      	| 1.95 	| 3.98 	| 531.2   	|
> | $z_G$ only      	| 1.88 	| 3.83 	| 530.5   	|
> | $z_G$ \& $z_A$	| **1.69** 	| **3.52** 	| **504.9** 	|
>
>
> |   Charged | ADE   	| FDE   	|
> |--------------	|:-------:	|:-------:	|
> |  $z_A$ only     	| 0.560 	| 1.144 	|
> | $z_G$ only     	| 0.541 	| 1.104 	|
> | $z_G$ \& $z_A$	| **0.52** 	| **1.09** 	|
>
>
> **Figure 4: The color of the ball (brown) is very difficult to distinguish given the background of the court.**
>
> Thanks for the suggestion. We will choose a less confusing color for the ball.
>
> **The explanation of Figure 4 at line 266 can be improved by outlining that with a fixed $z_G$ the ball always remains with agent 2. It's not entirely clear what happens when we fix $z_A$.**
>
> Here we show that multi-modality mainly comes from the inter-agent $z_G$, as fixing $z_A$ still resulted in multiple possible trajectories.
>
> **For the NBA dataset, is there a special identifier for the ball to differentiate it from players in the encoding? If not, how does the model know to treat it's movement differently?**
>
> The ball is treated as an agent just like other players. It is true that the ball’s trajectory can be very different from the players’. We let the model itself learn this distinction.
>
> **References 18 and 19 are the same.**
>
> Thanks for catching it. We will remove the duplicate.

---

> > ### Comment · Reviewer_8qCr · 2021-08-26
> > **Thanks for your response!**
> >
> > The authors addressed most of my concerns with the rebuttal and I am happy to see that they fixed a bug and improved their results. I ask that these clarifications and new results be carefully added to the final manuscript.

---

### Official Review · Reviewer_pxYX · 2021-07-17

**Rating:** 6
**Confidence:** 5

**Summary:**

This paper proposes to model interactions together with intention using the conditional generative models. For this, the authors created a model to encode two sets of latent variables and provided how they can be used for trajectory prediction. The evaluation is conducted using three datasets (two in the main manuscript and one in the supplementary).

**Limitations And Societal Impact:**

The authors provide a discussion on privacy-related societal impact. However, the authors should be more thoroughly discuss about it considering more broader aspects.

**Main Review:**

Strengths:
- The work is well-motivated, and introduction clearly describes it.
- The methods sounds technical.

Major concerns:
- One of contributions seems that it samples two types of latent variables, $z_G$ for interaction modeling and $z_A$ for intention modeling. However, I cannot find the validation of such categorization, except verbal description. I would like to see the influence of $z_G$ and $z_A$.
- In addition, the encoder of GRIN already extracts certain global features as a result of GCN in Eqn.2, considering the interactions of agents. In this view, it is questionable if it is good to categorize $z_G$ and $z_A$ into inter- and intra-agent latent. I don't see any intention information from the generation of $z_A$, and it seems more like spatial, temporal and inter-agent relations similar to [24,7].
- Following the previous concerns, one ablative study is that the demonstration of using two latent variables instead of one. Simply, the authors use $z_G,i$ in Eqn. 7. Since both $z_G$ and $z_A$ are sampled from the same latent space, I believe the result might not be huge different from such a replacement.
- It will be interesting to see the comparison of [24] in the experiments section in addition to [7,17], as it can be categorized as one of the closest work.
- The evaluation is made using two datasets. However, the Charged dataset provides simple toy examples, and it seems that the NBA dataset is tailored by authors. I am concerned about the validity of these evaluations. The readers of this paper would want to see more thorough comparisons with more diverse works (not only NRI-families but also recent methods) evaluated on more standard datasets. Although the authors provide the evaluation on one of the frequently used benchmark dataset (ETH/UCY) in the supplementary material, its evaluation criteria is different from the standard evaluation in literature, as they sample 100 trajectories. This also makes me doubt about the validity of this method.

Minor concerns:
- Figure 1 is missing the condition $X_p$ in the decoder.



-----------------
After read the authors' rebuttal, I raise my score.

**Time Spent Reviewing:**

2

---

> ### Author Response · Authors · 2021-08-10
> **Response to Reviewer pxYX**
>
> We thank the reviewer for the detailed review as well as the suggestions for improvement. Our responses to the reviewer’s comments are below.
>
> **I cannot find the validation of such categorization, except verbal description. I would like to see the influence of zG and zA.**
>
> To demonstrate the effects of $z_G$ and $z_A$, we perform a pair of interrogatory experiments on the Charged dataset:
> 1. We fix $z_{A,0},\dots,z_{A,4}$ for all five particles and $z_{G,0},\dots,z_{G,3}$ for four of the five particles, and only sample the $z_{G,4}$ for one particle, then observe the distribution of generated trajectories.
> 2. We fix the $z_{G,0},\dots,z_{G,4}$ for all five particles and $z_{A,0},\dots,z_{A,3}$ for four of the five particles, and only sample the $z_{A,4}$ for one particle, then observe the distribution of generated trajectories.
>
> In this pair of experiments, we observe:
> 1. When sampling only $z_{G,4}$ for one particle, the generated trajectories of all five particles are affected.
> 2. When sampling only $z_{A,4}$ for one particle, only its own generated trajectory is affected, and all other four particles' trajectories are not affected. This clearly shows the disentanglement of inter- and intra-agent factors.
>
> Please check the visualization through these anonymous links:
> 1. https://ibb.co/HpSP6gh
> 2. https://ibb.co/TWtN2xm
>
> In the visualization, the thick solid lines are ground-truth trajectories, and thin semi-transparent lines are generated trajectories from samples.
>
> **It is questionable if it is good to categorize $z_G$ and $z_A$ into inter- and intra-agent latent. I don't see any intention information from the generation of $z_A$**
>
> Please see the experiment described above for intention information. Conceptually, we refer to intention as features private to each agent. From eq(6), $z_G$ specifies the attention $\alpha$ that weights the ongoing interaction ($x_i^t$) through time between the agents. Had $\alpha$ contained primarily private information within the constrained softmax simplex, this information would be substantially diluted by the ongoing dynamic trajectory and the averaging in message-passing eq(6). In contrast, $z_A$ enters through eq(7) and directly affects the instantaneous displacement for each agent, so the model should retain individual intention in this variable. Importantly, it does not interact with individual trajectories of other agents $x_i^t$ but the combined message from the others $h_i^t$.
>
> In our work, $z_G$ and $z_A$ go through different generative processes at the decoder. One could also constrain the encoder to ensure intra- and inter-agent information are processed separately. But during the longer conditioning phase, the intention is likely to be affected by the paths of other agents. To ensure that the posterior is as accurate as possible, we do not constrain the encoder this way.
>
> **One ablative study is the demonstration of using two latent variables instead of one. Simply, the authors use $z_{G,i}$ in Eqn. 7. Since both $z_G$ and $z_A$ are sampled from the same latent space, I believe the result might not be hugely different from such a replacement.**
>
> We assume that the reviewer meant using one latent variable $z_G$ in this ablation. First, we clarify that $z_G$ and $z_A$ are in different subspaces, and only $z_G$ interacts between agents. This creates an inductive bias that encourages separate storage and processing of information. In situations where there is a separation between intention and interaction (e.g. NBA games), we may expect to see stronger transferability of our approach.
>
> We performed an extended ablation study inspired by the reviewer's suggestion. The architecture suggested is very close to but still worse than ours. In the table "single $z$" means the setting with only one $z$ vector, and the dimension of this $z$ vector is equivalent to the dimension of $z_G$ or $z_A$. The results are below.
>
> |    NBA          	| ADE   	| FDE   	| NLL     	|
> |--------------	|-------	|-------	|---------	|
> | single $z$		| 1.72 	| 3.62 	| 509.2 	|
> | $z_G$ \& $z_A$	| **1.69** 	| **3.52** 	| **504.9** 	|
>
>
> |   Charged | ADE   	| FDE   	|
> |--------------	|:-------:	|:-------:	|
> | single $z$		| 0.536 	| 1.137 	|
> | $z_G$ \& $z_A$	| **0.52** 	| **1.09** 	|
>
> Notice that the numbers for GRIN in the table below are better than the numbers in Table 1 of our paper submission. The improvement comes from these factors:
> 1. We fixed a few bugs that unfairly limited GRIN's performance in our submission, one example is that we didn't shuffle the data samples during the training of GRIN.
> 2. Inspired by reviewers C2Xv on the dimension of $z_A$ and $z_G$, we increase the dimension of $z_A$ and $z_G$ from 2 to 20.
>
> **It will be interesting to see the comparison of [24] in the experiments section in addition to [7,17], as it can be categorized as one of the closest work.**
>
> The code of EvolveGraph from [24] is not open source, which prohibits an empirical comparison.  Even so, although EvolveGraph is similar in that it tries to capture inter- and intra-agent effects, it remains distinct from ours in the following aspects:
> 1. The training objective: probabilistic (ours) vs error-based (theirs).
> 2. Their training procedure is not end-to-end and more customized.
> 3. They directly infer conditionally independent (factorized) discrete edge types, while we infer $z_G$ and then compute the real-valued interactions as an attention graph with advantages discussed in lines 132-140.
>
> **It seems that the NBA dataset is tailored by authors**
>
> Actually, we did not tailor the dataset in any way.  Rather, the NBA dataset is prepared by [29] and available at [this link](https://aws.amazon.com/marketplace/pp/prodview-7kigo63d3iln2). Also, the same dataset has been used and linked to in the [official repo](https://github.com/cgraber/cvpr_dNRI) of dNRI [7], which is one of our baselines. We did not do any further processing on this dataset.
>
> **Although the authors provide the evaluation on one of the frequently used benchmark dataset (ETH/UCY) in the supplementary material, its evaluation criteria is different from the standard evaluation in literature, as they sample 100 trajectories.**
>
> The ETH/UCY dataset we are using is prepared and uploaded by the authors of FQA [13] in the [official repo](https://github.com/nitinkamra1992/FQA/tree/master/data). The setting of using best-of-100 is introduced in several models having sampling capability to accommodate multi-modality. For example, Social-GAN [9] and Trajectron [11] baselines cited in our submission, also use this best-of-100 approach. To make the comparison fair, we did the same sampling strategy for all the baselines (NRI, dNRI, Social-GAN, Trajectron) having sampling capability.
>
> **Figure 1 is missing the condition Xp  in the decoder.**
>
> We will amend as suggested. Thanks.
>
> **The authors provide a discussion on privacy-related societal impact. However, the authors should more thoroughly discuss it considering more broader aspects.**
>
> We will do our best in addressing as many aspects of societal impact as possible. If GRIN is trained on a dataset that has an unbalanced number of samples from different groups, then the model is likely to reflect biases and misrepresentation of certain groups.

---

> > ### Comment · Reviewer_pxYX · 2021-08-24
> > **Thanks for the response**
> >
> > Thanks for proving the rebuttal. They mostly addressed my concerns and clarified. Please try add these clarifications and results in the final version as well as supplementary material, so the paper can be more solid.

---

### Decision · Program_Chairs · 2021-09-27

**Decision:**

Accept (Poster)

**Comment:**

This work explores two types of latent variables for multi-agent trajectory forecasting, an inter-agent latent code representing social relations and an intra-agent latent code representing agent intentions.   Scores for this paper were initially split between rejection and accept. Additional clarifications and (ablation) experiments provided during the rebuttal caused multiple reviewers to increase their scores.  While the experiments here are not extensive, they were perceived as being sufficient by the reviewers.  After the rebuttal period all reviewers recommend accepting this paper.

The AC recommends acceptance.